# Feedback from retinal ganglion cells to the inner retina

**Anastasiia Vlasiuk**[1,2], **Hiroki Asari** [1¤a¤b]*

**1** Epigenetics and Neurobiology Unit, EMBL Rome, European Molecular Biology Laboratory, Monterotondo, Rome, Italy, **2** Collaboration for joint PhD degree between EMBL and Heidelberg University, Faculty of Biosciences, Heidelberg, Germany

¤a Current address: Department of Molecular and Cellular Biology and Center for Brain Science, Harvard University, Cambridge, Massachusetts, United States of America
¤b Current address: Division of Biology and Biological Engineering, California Institute of Technology, Pasadena, California, United States of America
* asari@embl.it

**Data Availability Statement:** The data underlying this study is available on Zenodo (10.5281/zenodo. 5057577).

**Funding:** This work was supported by Gosney Postdoctoral Fellowship from CalTech and research grants from EMBL (H.A.). The funders

## Abstract

Retinal ganglion cells (RGCs) are thought to be strictly postsynaptic within the retina. They carry visual signals from the eye to the brain, but do not make chemical synapses onto other retinal neurons. Nevertheless, they form gap junctions with other RGCs and amacrine cells, providing possibilities for RGC signals to feed back into the inner retina. Here we identified such feedback circuitry in the salamander and mouse retinas. First, using biologically inspired circuit models, we found mutual inhibition among RGCs of the same type. We then experimentally determined that this effect is mediated by gap junctions with amacrine cells. Finally, we found that this negative feedback lowers RGC visual response gain without affecting feature selectivity. The principal neurons of the retina therefore participate in a recurrent circuit much as those in other brain areas, not being a mere collector of retinal signals, but are actively involved in visual computations.

## Introduction

Many brain circuits involve recurrent connections between principal neurons [1, 2]. This allows for feedback control that can shape the neuronal dynamics and plays critical roles in both local and global circuit functions, ranging from sensory information processing to decision making [3].

In the retina, the principal neurons—i.e., ganglion cells—are often thought to be strictly output elements [4, 5]. Previous studies have thus focused on characterizing how ganglion cells integrate signals from inner retinal neurons distributed in space and across different cell-types via chemical synapses [6–8]. As a result, two major mechanisms have been found that are important to control ganglion cell activity: feedforward signaling from bipolar cells to ganglion cells via amacrine cells [9–11], and feedback signaling between bipolar cells and amacrine cells [12–14].

Ganglion cells are, however, part of extensive gap junction networks in the retina [15–17]. While retinal neurons are generally coupled among those of the same type, ganglion cells also

had no role in study design, data collection and analysis, decision to publish, or preparation of the manuscript.

**Competing interests:** The authors have declared that no competing interests exist.

form electrical synapses with amacrine cells [18–20]. Therefore, ganglion cells could in principle signal back to the inner retina to modify the synaptic inputs they receive. Importantly, such feedback control of ganglion cell activity can be both excitatory and inhibitory. On the one hand, excitatory effects are expected from couplings through gap junction networks as they allow for electrical and bidirectional interactions among coupled neurons [16]. Different coupling architectures are indeed responsible for distinct firing patterns of ganglion cells over multiple time-scales. For example, direct couplings between ganglion cells underlie their nearly synchronous activity within a few milliseconds, whereas indirect couplings via amacrine cells mediate more broadly correlated activity of ganglion cells over tens to hundreds of milliseconds [17, 19–25]. On the other hand, inhibitory effects will be obtained if ganglion cells send feedback signals to amacrine cells through electrical synapses, while the amacrine cells in turn inhibit the ganglion cells or presynaptic bipolar cells through chemical synapses [26–29]. The computational power of ganglion cells should then be greater than is commonly thought, allowing for more complex and diverse visual processing. It remains unclear, though, if these feedback pathways are truly functional in the retina, and how they contribute to visual information processing in the retina.

Here we characterized such feedback circuitry in the salamander and mouse retinas by combining computational and experimental approaches. Using a biologically interpretable circuit model [30], we first show that ganglion cells exchange signals with each other in two distinct ways: faster, enhancing signals from proximal cells and slower, suppressing signals from distal cells. To experimentally test this model prediction, we then perturbed the ganglion cell activity by electrically stimulating the optic nerve emerging from an isolated retina, while simultaneously monitoring the consequences of the perturbation by multi-electrode array recordings. Consistently, the optic nerve stimulation produced a short period of enhanced firing (for tens of milliseconds), followed by a suppression of firing on longer time scales (for hundreds of milliseconds). The slower suppression was largely eliminated by pharmacologically blocking electrical or inhibitory synaptic transmission. Finally, by pairing the nerve shock with visual stimulation, we found that the negative feedback modulates the visual response gain but not the stimulus feature selectivity of ganglion cells. Therefore, we conclude that ganglion cells can actively control their activity via recurrent connections with amacrine cells, providing an additional mechanism for adaptive gain control in the retina.

## Results

### Circuit models with electrical couplings recapitulate retinal ganglion cell visual responses better than non-coupled models

How do ganglion cells interact with each other in the retina? To address this question, we started with a modeling approach that includes subthreshold effects of electrical couplings between retinal ganglion cells [15, 16]. Here we followed the cascade model framework and extended the neural circuit models of [30]: specifically, the full "linear-nonlinear-feedback-delayed-sum-nonlinear-feedback" (LNFDSNF) model as well as the reduced "linear-nonlinear-sum-nonlinear" (LNSN) model were employed as a baseline non-coupled model. Briefly, the first LNF stages collectively work as spatial subunits that correspond to bipolar cells and their upstream circuitry; the middle D stage is a delay in lateral propagation introduced by amacrine cells; and the last SNF stages represent the spatial summation by a target ganglion cell (see Methods for details).

Our coupled models (LNFDSCNF, S1A Fig; and LNSCN, Fig 1A) were then built by incorporating the following two parameters into the non-coupled models. The first parameter is the net coupling strength that represents the effect of both direct coupling between ganglion cells

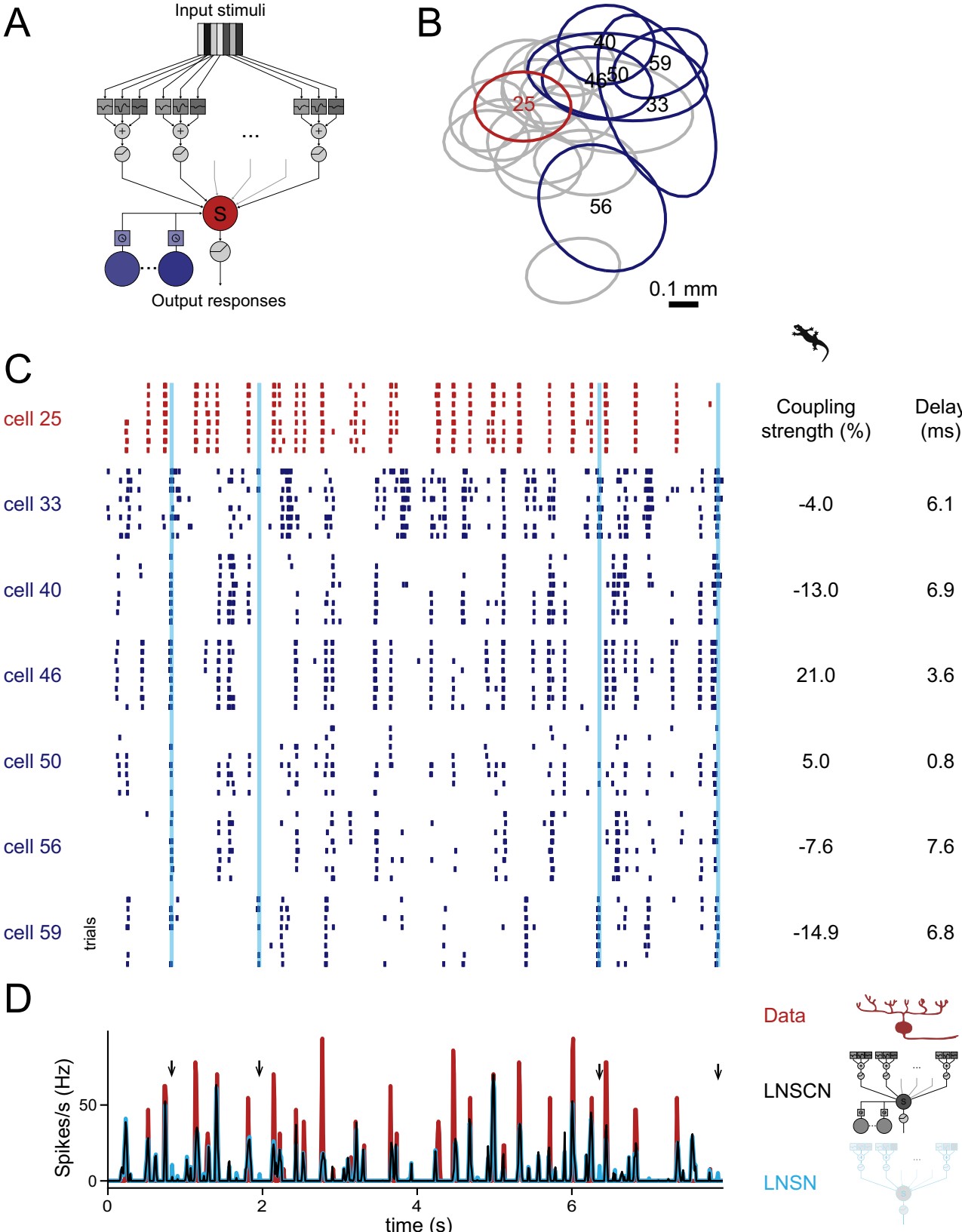

**Fig 1. Coupling model suppresses false-positive responses in non-coupling model.** (A) Schematic diagram of the circuit model with couplings among retinal ganglion cells ("LNSCN" model; see also S1 Fig). The target ganglion cell (red) integrates inputs from upstream bipolar cells as well as surrounding retinal ganglion cells (blue) with certain coupling strengths and delays. The model without couplings is identical to the "LNSN" model in [30]. (B) Receptive field centers of 20 OFF ganglion cells simultaneously recorded from an isolated salamander retina (red, representative modelled cell in C and D; blue, cells included in the coupling model for the representative cell; gray, cells excluded from the coupling model due to high spike correlations with the modelled cell; see also S2A Fig). Each outline represents a two-dimensional Gaussian fit to the receptive field profile (contour at 1 standard deviation). (C) Responses of ganglion cells (red, representative modelled cell from B; blue, cells included in the coupling model) to repetitions of the white noise stimulus (8 repeats in total). Each row in the raster denotes spikes from a single stimulus repeat for each cell. Cyan lines indicate timepoints where false positive responses in the non-coupled LNSN model were suppressed in the coupled LNSCN model (arrows in D). (D) Time course of the firing rate of the representative cell (red) and that of the model outputs with (black, LNSCN model) and without (cyan, LNSN model) couplings fitted to this cell. The LNSN model occasionally mis-predicted responses at which the cell did not fire spikes, but the LNSCN model correctly suppressed them in many cases (arrows). The coupling parameters (coupling strength and delay) for each surrounding cell included in the LNSCN model are shown next to the raster graph in C.

and indirect pathways via amacrine cells ($a_k$ in Eq (2) in Methods). A positive value is expected for a directly coupled pair of ganglion cells [23], where the coupling strength largely depends on the number of electrical synapses between the cells and their conductance [16]. In contrast, a negative value is expected for an indirectly coupled cell pair, where signals from one ganglion cell transmit to an amacrine cell via electrical synapses and the amacrine cell in turn inhibits the other ganglion cell via chemical synapses [26–29]. The second parameter is the net latency for signals to arrive from the neighboring ganglion cell to the modelled cell ($l_k$ in Eq (1) in Methods). A short latency is expected for a directly coupled pair, whereas a long latency is expected for an indirectly coupled pair. Importantly, here we kept the model size minimum to achieve reliable data fitting, with only two extra free parameters for each surrounding ganglion cell to describe the net effect of its action potentials on a target cell. Nevertheless, our coupled models are flexible enough to account for the overall effect of the interactions between retinal ganglion cells through the entire circuitry of the inner retina.

When fitted, these coupled and non-coupled models were both able to approximate the firing patterns of salamander [30–32] and mouse [33] retinal ganglion cells in response to white-noise visual stimuli consisting of randomly flickering bars (e.g., Fig 1B–1D; see Methods for details). The non-coupled models, however, occasionally predicted visual responses at times when they should not, while the coupled models often correctly suppressed such false-positive responses by the firing of negatively coupled cells that have the same response polarity as the target cell (e.g., Fig 1C and 1D for the reduced models; S1F and S1G Fig for the full models). This led to a small but statistically significant improvement of the coupled models in recapitulating the ganglion cell visual responses compared to the non-coupled models ($R^2 = 0.314 \pm 0.091$ versus $0.297 \pm 0.087$, $p<0.001$, for salamander cells, Fig 2A; $R^2 = 0.215 \pm 0.088$ versus $0.208 \pm 0.087$, $p<0.001$, for mouse cells, Fig 2C; mean ± standard deviation of the coefficient of determination (Eq (3) in Methods) for the reduced models, paired t-test; see also S1 Fig for the full models). In contrast, surrounding cells of the opposite response polarities contributed less to the performance of the coupled models ($R^2 = 0.322 \pm 0.095$ versus $0.309 \pm 0.010$, $p<0.001$, for salamander cells, Fig 2B; $R^2 = 0.209 \pm 0.090$ versus $0.208 \pm 0.087$, $p>0.5$, for mouse cells, Fig 2D). Importantly, to avoid confounding effects of common visual inputs in the coupled model, we considered only those surrounding cells that have low spike correlations—hence virtually no receptive field overlap—with the target cell (S2A and S2B Fig). Furthermore, we confirmed the convergence and robustness of the model fitting by the perturbation analysis where we reoptimized the models after resetting the coupling parameters (S2C and S2D Fig; see Methods for details). Therefore, as suggested by anatomical studies [18], our circuit model analysis supports the presence and solid contributions of couplings between retinal ganglion cells, especially those of the same response polarity, to shaping their visual response dynamics.

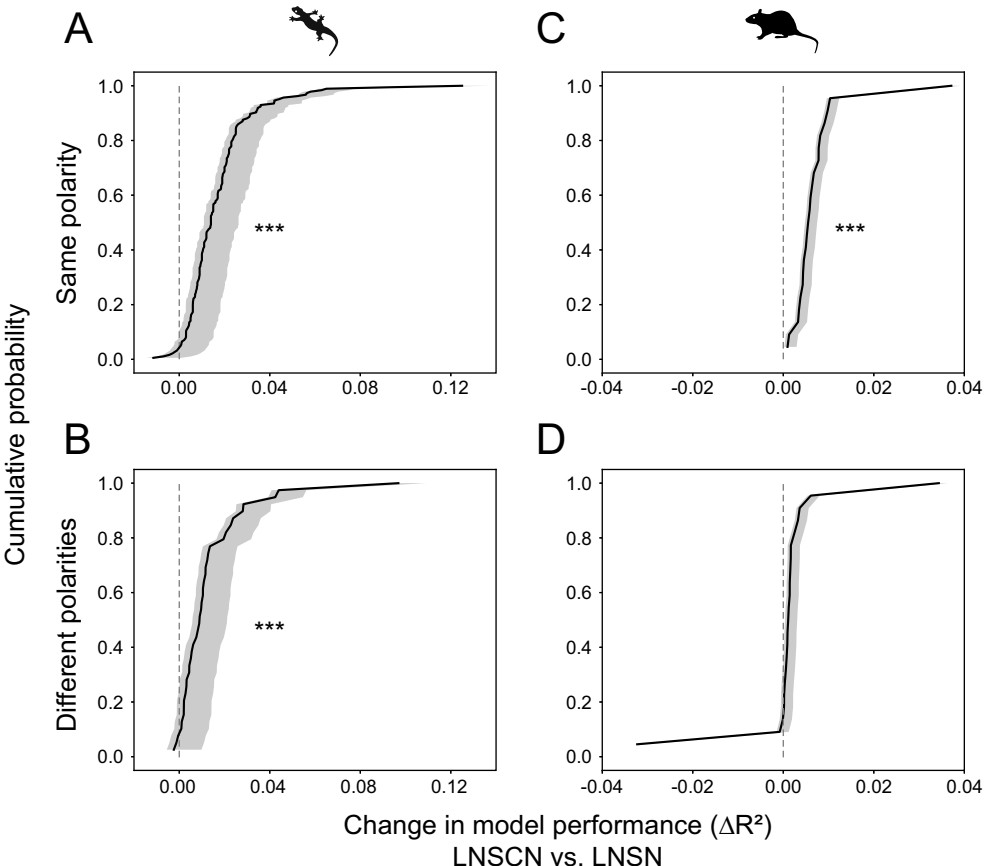

**Fig 2. Coupling model outperforms non-coupling model.** Cumulative probability distributions (black; confidence intervals at 5 and 95 percentiles from perturbation analysis in gray; see Methods for details) of the LNSCN model performance gain over LNSN model for salamander ganglion cells, calculated as a difference of the coefficients of determination ($R^2$, Eq (3) in Methods; see also S1 Fig). (A, B) The LNSCN model outperforms the LNSN model more with the neighboring cells of the same response polarity (A; $N$ = 185 OFF cells with OFF neighbors; $\Delta R^2$ = 0.017±0.016, mean ± standard deviation; $p < 0.001$, paired t-test) than with those of different polarities (B; $N$ = 35 OFF cells with ON neighbors and 4 ON cells with OFF neighbors; $\Delta R^2$ = 0.013±0.017; $p < 0.001$): $p$ = 0.006 from Kolmogorov-Smirnov test. Here and thereafter, three stars (★★★) indicate $p < 0.001$; ★★, $p < 0.01$; and ★, $p < 0.05$. (C, D) Corresponding figure panel for the mouse ganglion cells. The LNSCN model outperforms the LNSN model when the neighboring cells of the same response polarity are included (C; $N$ = 10 ON cells with ON neighbors and 12 OFF cells with OFF neighbors; $\Delta R^2$ = 0.007±0.007; $p < 0.001$), but not with the neighboring cells of different polarities (D; $N$ = 10 ON cells with OFF neighbors and 12 OFF cells with ON neighbors; $\Delta R^2$ = 0.001±0.010; $p > 0.5$): $p < 0.001$ from Kolmogorov-Smirnov test.

## Circuit model analysis predicts both positive and negative couplings between retinal ganglion cells

We next analyzed the model parameters to gain insights into the retinal circuits underlying ganglion cell couplings and to derive experimentally testable predictions. The data sets from ON and OFF cells were combined here as they showed the same trend (Fig 3 and S3 Fig).

First, we found both positive and negative couplings between ganglion cells of the same response polarities (Fig 3A and 3B and S3A and S3B Fig for salamander retinas; Fig 3E and 3F and S3E and S3F Fig for mouse retinas). A large fraction of these cells (salamander, 26%; mouse, 55%) received both positive and negative couplings (e.g., Fig 1C). Moreover, these couplings were frequently reciprocal, either mutually positive or negative between given cell pairs ($p < 0.001$, $\chi^2$-test; Fig 3G and 3I and S3G and S3I Fig). These indicate that a single ganglion cell can be involved in multiple feedback pathways, imposing distinct effects on different

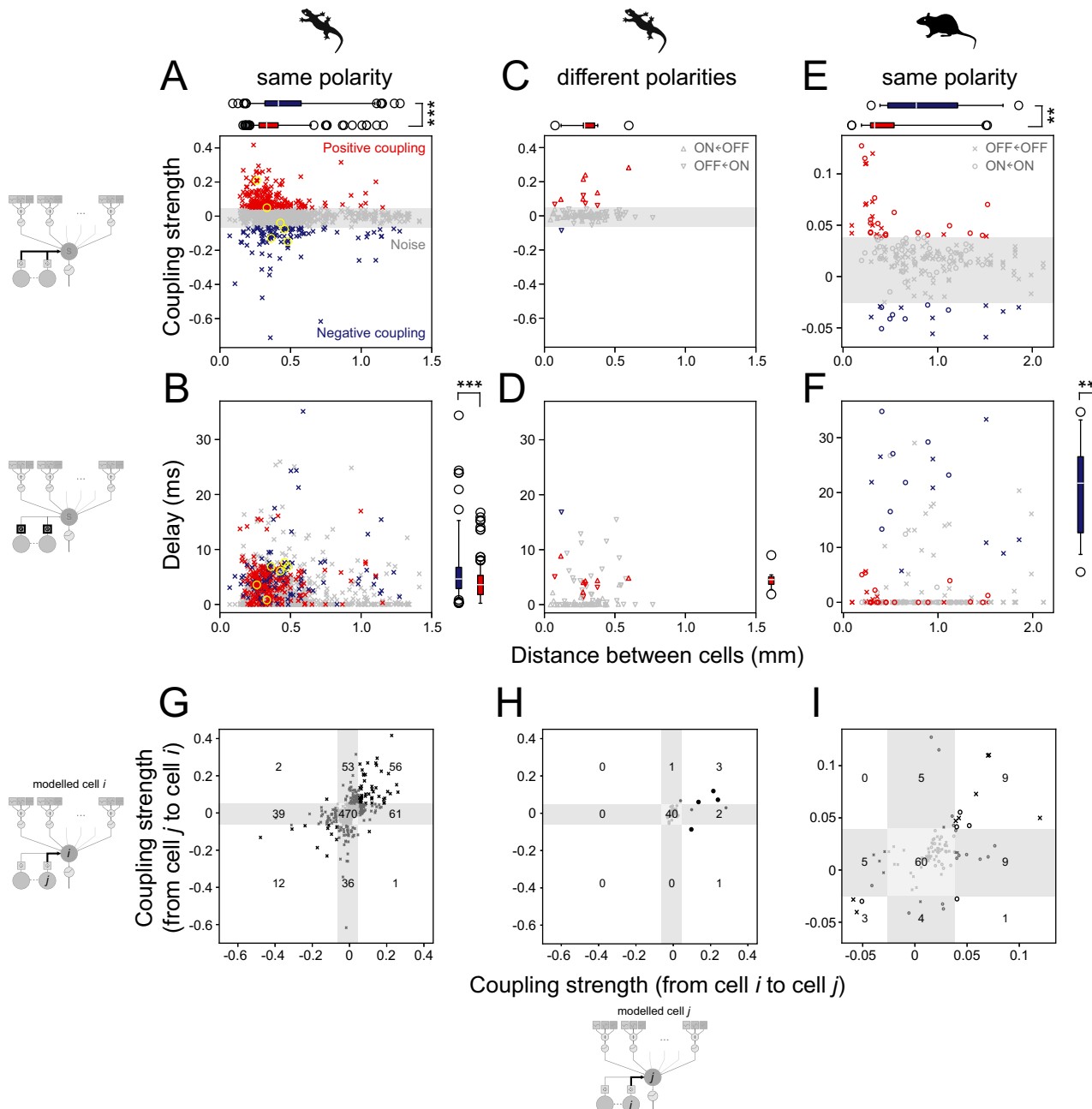

**Fig 3. Coupling model predicts faster enhancing effects among proximal retinal ganglion cells and slower suppressing effects among distal cells.**
(A, B) Coupling strength (A; $a_k$ in Eq (2) in Methods) and delay (B; $l_k$ in Eq (1) in Methods) parameter values from the LNSCN model plotted as a function of the distance between salamander retinal ganglion cells of the same response polarity (crosses, OFF cells coupled with OFF neighbors, $N = 1460$; yellow circles, representative cases in Fig 1B–1D). The noise level was determined by shuffling analysis (gray, 0.5 and 99.5 percentile; see Methods for details). Positive couplings (red, $N = 211$) were found at a shorter distance (A; 0.31±0.14 mm versus 0.40±0.26 mm; median ± interquartile range as shown by the box plot; $p < 0.001$, rank-sum test) with a shorter latency (B; 3.5±3.6 ms versus 4.6±4.0 ms; $p < 0.001$) than negative couplings (blue, $N = 98$). (C, D) Corresponding figure panels for salamander cells with different response polarities (down-pointing triangles, OFF cells with ON neighbors, $N = 47$; up-pointing triangles, ON cells with OFF neighbors, $N = 46$). Couplings above the noise level were mostly positive ($N = 10$) with the distance of 0.28±0.08 mm (C; median ± interquartile range) and the latency of 4.2±1.5 ms (D). (E, F) Corresponding figure panels for mouse retinal ganglion cells of the same response polarity (circles, ON cells coupled with ON cells, $N = 76$; crosses, OFF cells coupled with OFF cells, $N = 116$). Positive couplings (red, 18 ON and 15 OFF cell pairs) were found at a shorter distance (C; 0.33±0.25 mm versus 0.78±0.74 mm; $p = 0.002$) and with a shorter latency (D; 0.0±0.0 versus 21.9±13.8 ms; $p < 0.001$) than negative couplings (blue, 7 ON and 9 OFF cell pairs). Data not analyzed for the mouse cells with different response polarities because the coupled model performance did not show an improvement (Fig 2B). (G–I) Comparison of the coupling strengths from one cell to another and vice versa (black, couplings above the noise level in both directions; dark grey, above the noise level only in one direction; light gray, both below the noise level). The number of data points in each category is shown in the figure panels. Symmetric, either mutually

positive or negative, couplings were found more frequently than expected in both salamander (G, between cells of the same response polarity; H, between cells with different response polarities) and mouse (I, between cells of the same response polarity) retinas ($p<0.001$ in all three cases, $\chi^2$-test with $df=4$). In G and I, crosses indicate OFF cell pairs, and circles ON cell pairs. In H, each data point represents an ON-OFF cell pair.

surrounding cells. In contrast, positive couplings dominated between the cell pairs with different response polarities (Fig 3C, 3D and 3H and S3C, S3D and S3H Fig for salamander cells; mouse data not analyzed because the coupled model performance did not improve significantly; Fig 2D and S1E Fig). Consistently, such a cross-talk between the ON and OFF channels via gap junctions has been reported recently [34]. Taken together, negative couplings are suggested to be formed exclusively between ganglion cells of the same response polarities.

Second, positive couplings were on average found between cell pairs at a shorter distance and had a shorter latency than negative couplings (Fig 3A–3F and S3A–S3F Fig). Given that the diameter of ganglion cell dendritic fields is around 0.3 mm [18, 35], the distance of the positive couplings (0.31±0.14 and 0.33±0.25 mm for salamander and mouse cell pairs, respectively, median ± interquartile range from the reduced models) is consistent with direct interactions between ganglion cells, although this does not exclude possible contributions of indirect electrical couplings via amacrine cells [20]. In contrast, the distance of the negative couplings (0.40±0.26 and 0.78±0.74 mm, respectively) exceeds what one expects for direct coupling. The circuit model thus predicts that faster enhancing effects between proximal ganglion cells can arise from direct coupling, whereas slower suppressing effects between distal cells should be mediated by indirect coupling, presumably via inhibitory amacrine cells in the inner retina [27–29]. The prediction on the positive couplings is well supported by previous studies [17, 19, 23, 36, 37], but direct experimental evidence is still lacking on the slow negative coupling between ganglion cells over a long distance.

Third, there was no marked difference in the common parameters between coupled and non-coupled models (S4 Fig). Consistently, the outputs of these two models showed nearly identical dynamics (e.g., Fig 1D), suggesting that the ganglion cell couplings have little effect on the cell's stimulus feature selectivity. Importantly, however, non-desired response peaks were present more frequently in the output of the non-coupled models than that of the coupled models (e.g., Fig 1D). This indicates that the ganglion cell couplings have some transient effects on the response gain of a target cell, suppressing such false-positive responses that were present otherwise.

In summary, our circuit model analysis predicts that, besides bidirectional excitation via gap junction networks [16], retinal ganglion cells of the same response polarity can have reciprocal indirect inhibition between each other for a transient gain control from outside their receptive fields.

## Optic nerve stimulation imposes diverse effects on ganglion cell activity

To directly test the model prediction of an indirect inhibition between retinal ganglion cells (Figs 1–3 and S1–S4 Figs), we next experimentally examined how the firing patterns of ganglion cells are affected by their own action potentials and the action potentials of other ganglion cells. For going beyond correlation analysis [27], here we sought to perturb the activity of many ganglion cells simultaneously to strongly drive the recurrent signaling pathways, but in a manner independent of light signaling pathways. Using a suction electrode [38], we thus electrically stimulated the tip of the optic nerve (around 5 mm long) emerging from an isolated retina in the dark, and simultaneously recorded the spiking activity of ganglion cells to monitor the outcome of the perturbation (Fig 4A; see Methods for details). We then focused on those cells with sufficiently high baseline firing rates (>1 Hz) to reliably detect a change in

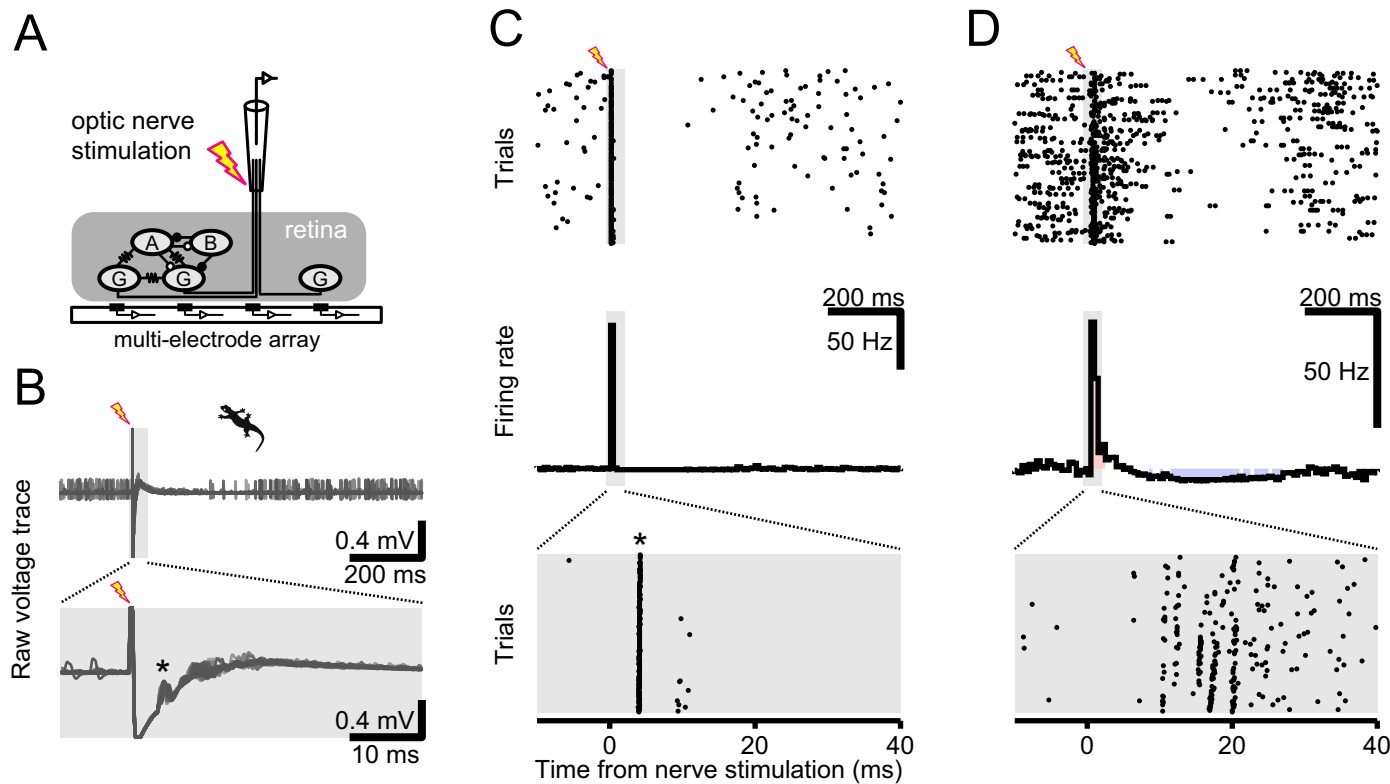

**Fig 4. Optic nerve stimulation produces fast excitation and slow inhibition in retinal ganglion cell firing activity.** (A) Schematic diagram of the experiment. The tip of the optic nerve emerging from an isolated retina was electrically stimulated with a suction electrode (inter-stimulus interval, 1–1.5 s), while a population of ganglion cells was simultaneously recorded with a multi-electrode array (see panel B for example). A, amacrine cell; B, bipolar cell; G, ganglion cell; open circle, inhibitory synapse; closed circle, excitatory synapse; resistor symbol, electrical synapse. The other retinal cell types and synapses are omitted for simplification. (B) Example raw data traces from an isolated salamander retina overlaid across trials of the optic nerve stimulation in the dark. The lightning symbol indicates the stimulation onset, and the gray area indicates the window for which the magnified traces are shown at the bottom panel. The antidromic spikes are indicated by the asterisk. (C, D) Representative responses of retinal ganglion cells to the optic nerve stimulation (top, raster graphs; bottom, peri-stimulus time histogram (PSTH), magnified at the bottom for the window indicated by the gray area). A period of suppression was observed in cells either with (C) or without (D) antidromically evoked spikes (asterisk). Red- and blue-shaded bins in the PSTHs indicate those in which the firing rate significantly increased or decreased from the spontaneous activity, respectively (significance level of 0.05 without correction by the number of time bins).

their activity. In particular, a decrease of the firing rates should follow the nerve stimulation if ganglion cells are part of the proposed negative feedback circuitry, whereas no significant change should be observed if the stimulated cells are all purely postsynaptic within the retina.

We found that the optic nerve stimulation had pronounced effects on the spontaneous activity of ganglion cells in both salamander and mouse retinas (Figs 4 and 5 and S5 Fig). About a third of the recorded cells (75 out of 193 salamander cells, 39%; 41 out of 132 mouse cells, 31%) showed immediate antidromically-evoked spikes after the nerve shock (e.g., Fig 4B and 4C and S5A and S5B Fig). In both cells with and without the antidromic spikes, we frequently observed a period of enhanced firing over tens of milliseconds (e.g., Fig 4D and S5B Fig) and a suppression of firing on a longer time scale over hundreds of milliseconds (e.g., Fig 4C and 4D and S5A and S5B Fig). The proportion and time course of these indirect enhancing and suppressing effects varied greatly between ganglion cells (Fig 5A and S5C Fig). Importantly, these indirect effects of the nerve stimulation were observed regardless of the presence of the antidromic spikes (e.g., Fig 4D). Moreover, applying the electrical stimulation to the bath without holding the optic nerve did not affect the ganglion cell spontaneous activity (3

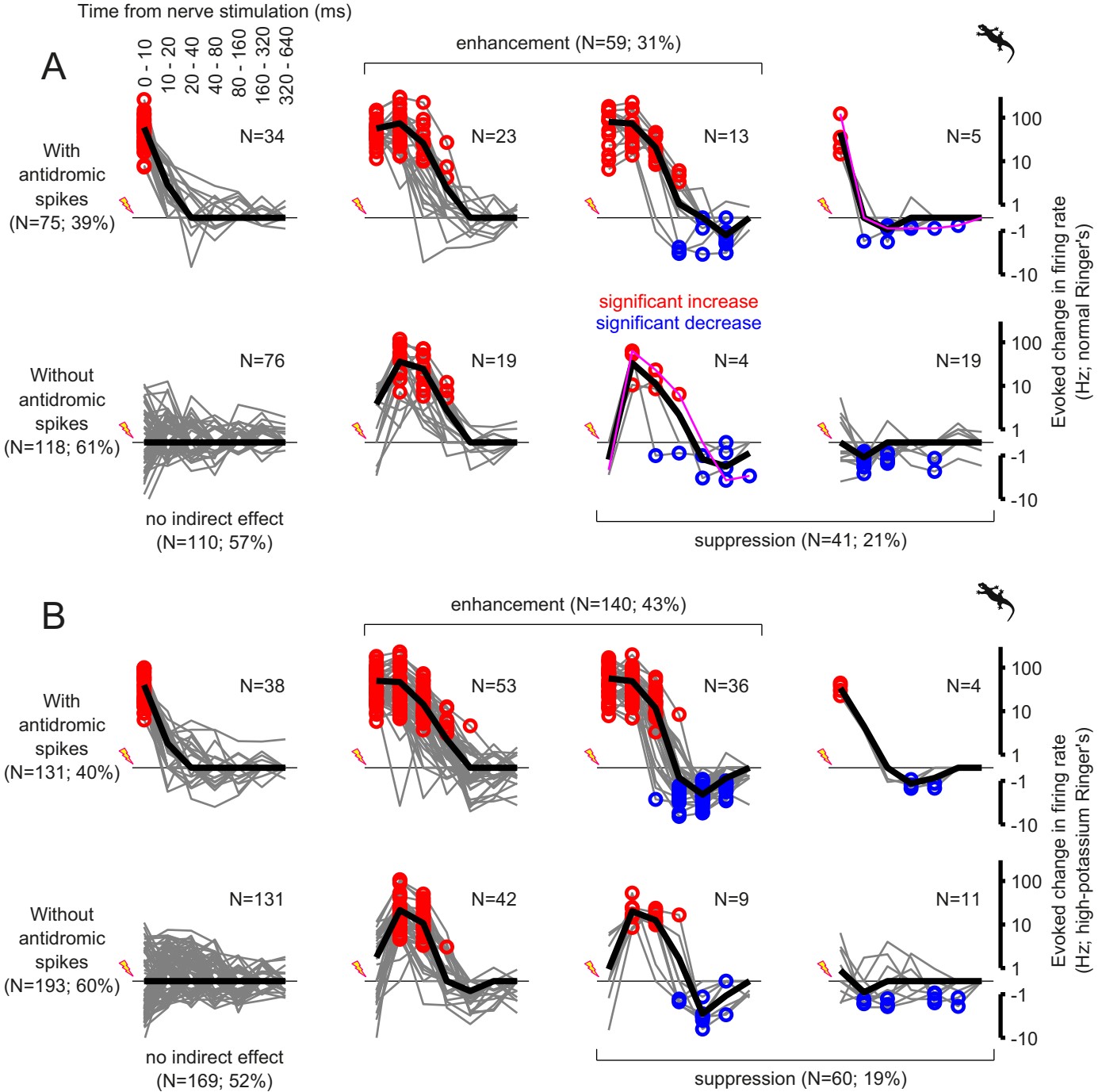

**Fig 5. Intrinsic cellular property affects the fast excitation but not slow inhibition after the optic nerve stimulation.** Population data of the salamander retinal ganglion cell responses to the optic nerve stimulation in the normal (A) and high-potassium (B; 13 mM KCl) Ringer's media, in the format of 2-by-4 contingency table. Each gray line represents the evoked change in the firing rate of a cell from its baseline (A, $N$ = 193 in total; B, $N$ = 324; red and blue circles, bins with significant increase and decrease, respectively, with Bonferroni correction), and the black line shows the mean over the cells in each group. In rows, the data were categorized by the presence (top; A, $N$ = 75, B, $N$ = 131; $p$ = 0.5, $\chi^2$-test) and absence (bottom) of antidromic spikes. In columns, the data were categorized by the outcome of the nerve stimulation: first column, no indirect effect; second and third columns, enhanced firing after the nerve stimulation (A, $N$ = 59; B, $N$ = 140; $p$ = 0.009); third and fourth columns, suppressed firing after the nerve stimulation (A, $N$ = 41; B, $N$ = 60; $p$ = 0.8). The representative responses in Fig 4C and 4D are shown in magenta in A. The cut-off firing rate for the log-PSTH is ±0.5 Hz.

salamander and 2 mouse retinas). Therefore, the observed changes in the firing after the nerve stimulation were likely mediated by signaling through circuits.

To examine how the intrinsic properties of the cells affect the outcome of the optic nerve stimulation, we repeated the experiments using a high-potassium Ringer's medium (13 mM KCl replacing NaCl; Fig 5B). In this condition, ganglion cells generally showed an increased spontaneous activity (from 0.7±1.6 Hz to 2.0±3.2 Hz, mean ± standard deviation; $p<0.001$, sign-test; $N$ = 855 salamander cells). Nevertheless, there was no change in the fraction of the cells with antidromic spikes (control, 39% (75 out of 193 cells), Fig 5A; high-potassium, 40% (131 out of 324 cells), Fig 5B; $p$ = 0.5, $\chi^2$-test), validating the reliability of our experimental procedures. The fraction of the cells with a suppression after the nerve shock remained unchanged as well (21% (41 out of 193 cells) versus 19% (60 out of 324 cells); $p$ = 0.8). This supports little contribution of the cell intrinsic mechanisms to the suppression, such as afterhyperpolarization [39]. In contrast, the fraction of the cells with an enhanced firing increased significantly (from 31% (59 out of 193 cells) to 43% (140 out of 324 cells); $p$ = 0.008). This indicates that the fast enhancement of the ganglion cell activity after the nerve shock was susceptible to a change in the cell intrinsic properties, and thus cannot be attributed solely to the network effects as we cannot exclude possible changes in the ganglion cell properties due to the nerve stimulation.

Taken together, consistent with our model predictions, these data provide direct evidence that ganglion cells can drive negative feedback signaling in the inner retina besides previously described excitatory interactions via gap junction networks [17, 19–25].

## Recurrent inhibitory circuits from retinal ganglion cells involve gap junctions and amacrine cells

What are the retinal circuits underlying such recurrent inhibitory signaling from ganglion cells? Previous anatomical and computational studies suggest that ganglion cells can form a recurrent circuit with amacrine cells via gap junctions [16, 18, 26]. Here we took a pharmacological approach in the salamander retina to test this circuit hypothesis. We first blocked gap junctions by applying 100 μM meclofenamic acid (Fig 6A and 6B). This had little effect on the fast enhancement of the firing (19 out of 19 cells; $p<0.001$, Fisher's exact test), indicating that the optic nerve stimulation itself can affect the intrinsic excitability of the cells for a short period of time beyond driving the antidromic spikes (see also Fig 5). Nevertheless, the drug application generally led to an elimination of the slow suppression after the nerve shock in the dark (7 out of 8 cells; $p$ = 0.036, Fisher's exact test). Electrical synapses were thus found indispensable for the negative feedback signaling from ganglion cells. To test the involvement of amacrine cells in this signaling pathway, we next blocked γ-aminobutyric acid (GABA) and glycine transmission by applying 100 μM picrotoxin and 1.0 μM strychnine (Fig 6C and 6D). We found that these inhibitory transmission blockers also abolished the slow suppression after the nerve stimulation in the dark (7 out of 7 cells; $p$ = 0.008, Fisher's exact test), while the fast enhancement remained intact (13 out of 17 cells; $p$ = 0.006, Fisher's exact test). Taken together, these results support that the negative feedback pathway is physiologically functional from ganglion cells to amacrine cells via gap junctions [26–29].

## Negative feedback signaling modulates gain but not selectivity of ganglion cell visual responses

Thus far we have electrophysiologically examined the negative feedback without presenting visual stimuli. But, how does it affect the visual response properties of retinal ganglion cells? Our model predicts a role in transiently controlling the response gain (Fig 1D). To experimentally test this, we next presented white-noise visual stimuli in combination with the optic nerve

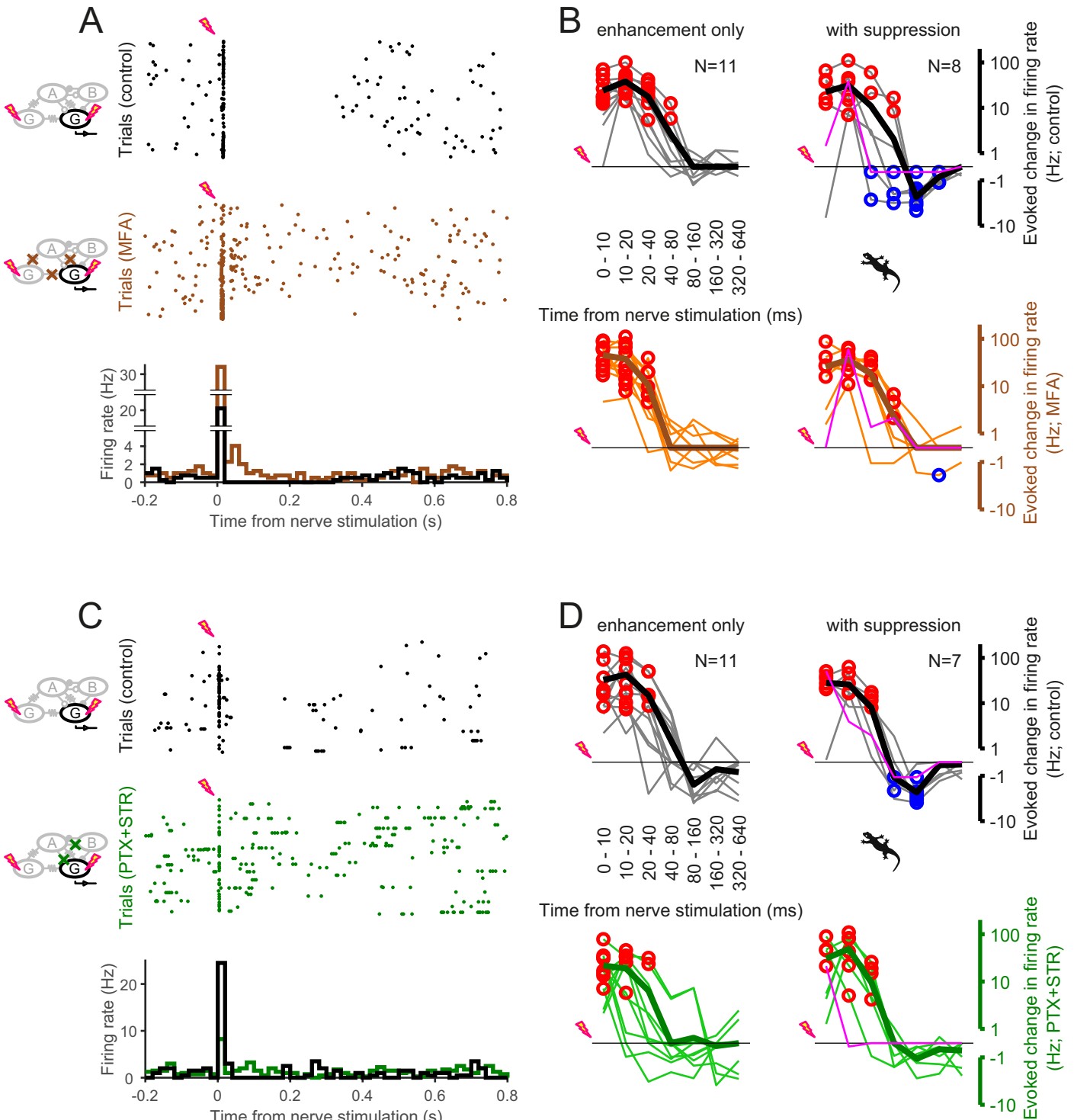

**Fig 6. Negative feedback involves electrical synapses and inhibitory synaptic transmission.** (A) Spiking activity of a salamander retinal ganglion cell in the dark (top, raster graphs; bottom, PSTHs) in response to the optic nerve stimulation in the absence (black; control) and presence (brown) of gap junction blockers (100 μM meclofenamic acid; MFA). (B) Summary of the effects of blocking gap junctions on the ganglion cell firing patterns after the nerve stimulation (top, control; bottom,

with MFA), shown in the same format as Fig 5 (magenta, the representative cell in A). The suppression after the nerve stimulation was abolished after blocking electrical synapses (right: 7 out of 8 cells; $p = 0.036$, Fisher's exact test), whereas the enhancement remained (left and right altogether; 19 out of 19 cells; $p<0.001$). (C, D) A representative example (C; in the same format as A) and the summary (D; in the same format as B) of the effects of blocking inhibitory synaptic transmission (100 μM picrotoxin and 1.0 μM strychnine; PTX+STR, green) on the ganglion cell firing patterns after the nerve stimulation. The suppression after the nerve stimulation was abolished after blocking inhibitory transmission (right: 7 out of 7 cells; $p = 0.008$), whereas the enhancement remained (left and right altogether except for the representative cell in magenta; 13 out of 17 cells; $p = 0.02$).

stimulation, and exploited the reverse-correlation methods to systematically examine changes in the response gain and feature selectivity of individual cells due to the nerve stimulation (Fig 7A and S6A Fig; [40, 41]). Specifically, we fitted a linear-nonlinear (LN) model to the visual responses of each cell at different time points from the nerve shock, and assessed i) the feature selectivity by an ON-OFF index (e.g., Fig 7C and 7D and S6C and S6D Fig), defined as the difference of the peak and valley values of the linear filter, normalized by the sum of the two (see Eq (4) in Methods for details) and the spectral peak frequency of the linter filter (e.g., Fig 7E and 7F and S6E and S6F Fig); and ii) the response gain by a sigmoid function fitted to the profile of the static nonlinearity from the LN model (Eqs (5) and (6) in Methods; e.g., Fig 7G and 7H and S6G and S6H Fig). In the present experiments, due to the presence of the stimulation pipette, we could not avoid a distortion of the visual stimuli projected from above the retina. Our analysis is thus limited to the full-field response properties analyzed with the LN model, but not extended to the spatiotemporal properties that could be analyzed with an LNSN model and beyond.

During a period of suppressed firing after the nerve shock (Fig 7B and S6B Fig), the linear filter profile remained largely unchanged (change in the ON-OFF index, −0.00±0.09, mean ± standard deviation, $p = 0.5$, sign-test, $N = 25$ salamander cells, Fig 7I; change in the spectral peak frequency, 0.01±0.46 Hz, $p = 0.8$, Fig 7J; see also S6I and S6J Fig for the mouse cells). In contrast, it sometimes showed a marked change during a period of enhanced firing: e.g., from monophasic (having one positive or negative phase) to biphasic (having both positive and negative phases; S6C Fig). Nevertheless, both measures of the feature selectivity did not show significant changes across the population (0.02±0.14 and −0.00±1.18 Hz for the ON-OFF index and spectral peak frequency, respectively; both with $p>0.3$; $N = 95$ salamander cells). These data suggest that the recurrent signals from ganglion cells had little effect on the feature selectivity of their visual responses.

The response gain, in contrast, was affected during both periods of enhancement and suppression after the nerve stimulation (Fig 7G, 7H, 7K, 7L and S6G, S6H, S6K, S6L Fig). On the one hand, the static nonlinear gain function was down-regulated during a period of suppression (e.g., Fig 7G and 7H and S6G and S6H Fig). In particular, the upper bound spike probability showed a significant decrease over the population (−0.03±0.02; $p<0.001$, sign-test, Fig 7K; see also S6K Fig) while the lower bound remained the same (−0.00±0.00; $p>0.5$; Fig 7L; see also S6L Fig). On the other hand, the gain was generally up-regulated during a period of enhancement (e.g., S6G and S6H Fig) with marginally increased upper bounds (0.03±0.08, $p = 0.08$, Fig 7K; see also S6K Fig) and significantly increased lower bounds (0.06±0.08, $p<0.001$, Fig 7L; see also S6L Fig) over the population. It should be noted, though, that this non-specific increase in the activity resulted not only from the gap junction network effect, but also from a possible brief change in the cell's excitability due to the nerve stimulation (see also Figs 5 and 6).

Taken together, these results indicate that the negative feedback signals primarily contribute to modulating the response gain, while ganglion cells may adaptively change their visual response properties by exploiting the whole recurrent circuitry.

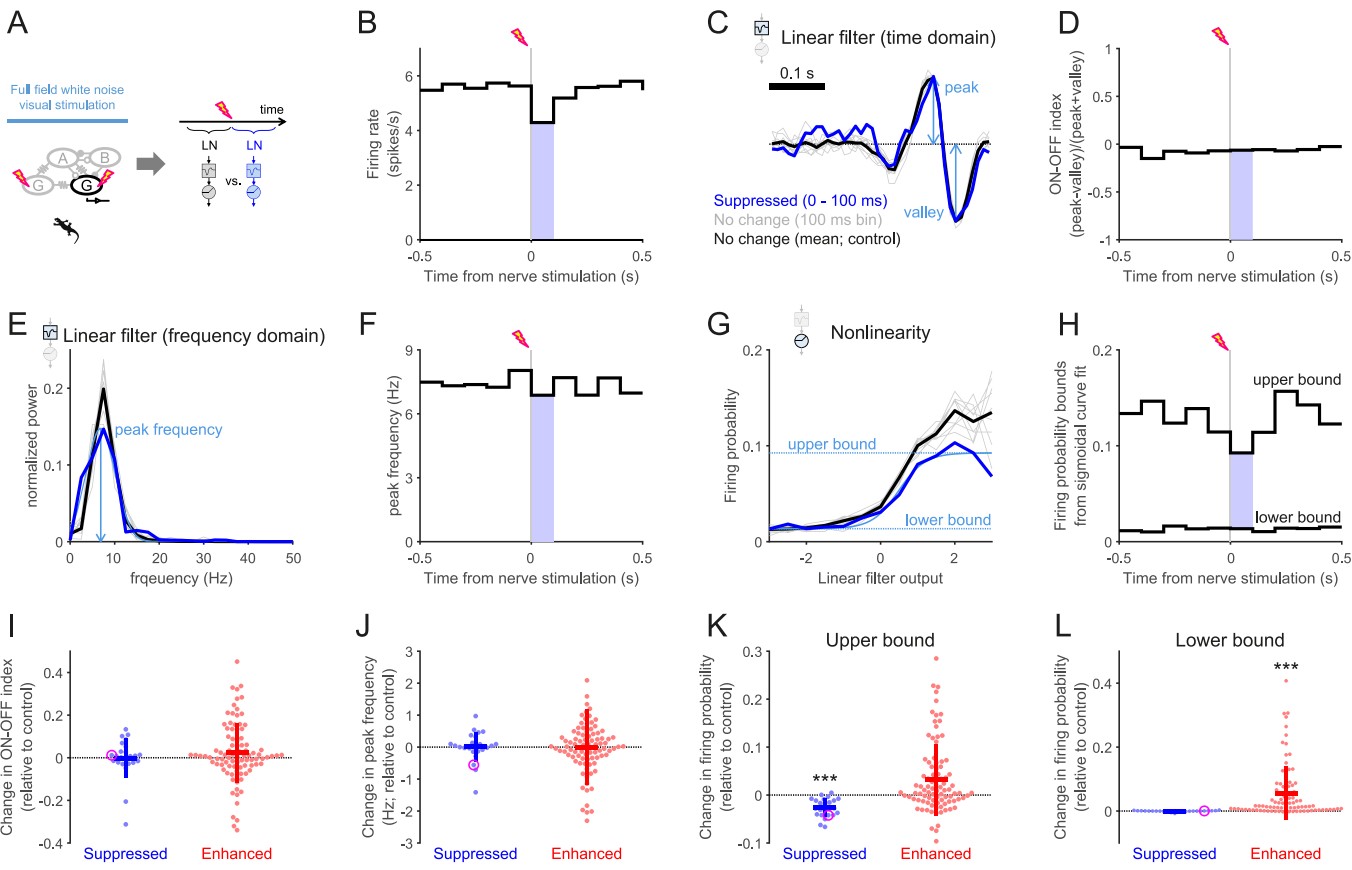

**Fig 7. Negative feedback modulates the visual response gain but not the feature selectivity.** (A) Schematic diagram of the experiment and analysis. Full-field white-noise visual stimuli were presented together with the optic nerve stimulation (inter-stimulus intervals, 1–1.5 s) to probe changes in the LN model parameters at different time windows from the nerve stimulation (see B–H for example). The linear filter represents the stimulus feature selectivity of a cell, while the static nonlinearity indicates the visual response gain. (B) PSTH of a representative salamander ganglion cell with respect to the optic nerve stimulation during the visual stimulus presentation. Blue-shaded bins (0–100 ms after the nerve shock) indicate those in which the firing rate was significantly lower than the baseline response of the cell (500 ms window before the nerve stimulation). (C) Linear filters of the example cell obtained by the reverse-correlation methods (i.e., spike-triggered average stimulus) using the spikes in different time bins from the PSTH (748–1037 spikes/bin; blue, 0–100 ms bin with significantly lower firing rates from B; gray, all the other 100 ms bins without significant firing rate changes; black, baseline). (D) Polarity of the linear filters (C) quantified by an ON-OFF index (i.e., the difference between the peak and valley values divided by the sum of the two; Eq (4) in Methods). (E) Normalized power spectral density of the linear filters in C. (F) Peak frequencies of the linear filters, estimated by fitting a gaussian curve to the power spectra in E. (G) Static nonlinearity of the example cell, computed for each corresponding linear filter at different time bins (Eq (5) in Methods). A sigmoid function (Eq (6) in Methods) was fitted to quantify the nonlinear profile (e.g., light blue for the suppressed period at 0–100 ms after the nerve shock, with the upper and lower bounds of the cell's firing probability in dotted horizontal lines). (H) The upper and lower bound dynamics of the example cell's firing probability with respect to the time from the nerve shock. The cell's response gain (upper bound) became lower during the period of suppressed firing, while the stimulus feature selectivity remained unchanged (D and F). (I, J) Summary of the changes in the ON-OFF indices (I) and the spectral peak frequencies (J) between the periods with and without significant firing rate changes after the nerve stimulation (blue, decrease N = 25; red, increase N = 94; magenta, the representative cells in B–H). The horizontal and vertical lines represent the mean and standard deviation, respectively. On average there was no significant difference from the baseline across population ($p>0.3$ for both cases in I and J, sign-test). (K, L) Summary of the changes in the sigmoid function parameters fitted to the nonlinearity. The upper bounds (K) significantly decreased during the period of suppression (blue, $p<0.001$, sign-test), but only marginally increased during the period of enhancement (red, $p = 0.08$). In contrast, the lower bounds (L) significantly increased during the enhancement ($p<0.001$) but remained the same during the suppression ($p>0.5$).

## Discussion

Circuit modeling is a powerful approach to integrate brain anatomy and physiology for better understanding the overall function of the system [42]. By explicitly representing individual neurons and their connections with the model parameters, circuit models can provide predictions on the structure and function of the target neuronal circuitry, and subsequently these

predictions can be tested by experiments [30]. Here we took this theory-driven approach to functionally characterize the ganglion cell feedback circuits in the inner retina. We first extended a retinal cascade model [30] to incorporate interactions between ganglion cells as suggested by anatomical studies [16, 18], and derived a prediction that retinal ganglion cells of the same response polarity form reciprocal inhibition over a long distance for a transient gain modulation (Figs 1–3 and S1–S4 Figs). We then experimentally validated that ganglion cells can indeed suppress the firing among themselves by propagating signals to amacrine cells via gap junctions (Figs 4–6 and S5 Fig). Furthermore, as predicted by our circuit model, we showed that this negative feedback lowers the visual response gain without much affecting the stimulus feature selectivity of the cells (Fig 7 and S6 Fig). The discovery of this new gain control mechanism in the retina not only highlights the importance of gap junctions in visual processing, but also promises that theory-driven approaches will further reveal neural circuit functions in future studies.

Retinal gap junction networks have been well conserved across species over evolution [25]. In both salamander and mouse retinas, our circuit model analysis indeed showed the presence of negative couplings among some distal ganglion cells, whereas positive couplings were found more frequently among proximal cells (Fig 3 and S3 Fig). This spatial organization is consistent with previous studies in both salamander [37] and primate retinas [27, 36]. Moreover, ganglion cells in both salamander and mouse retinas showed a period of suppression following the optic nerve stimulation under the ex vivo condition (Figs 4, 5 and S5 Fig). Such suppression was also reported in the catfish retina [28, 29] as well as in the primate retina [39], although the time scale in these previous reports was shorter than that in this study. Nevertheless, later modelling [26] and spike correlation studies [27] support that such suppression arises from the recurrent circuitry involving amacrine cells via gap junctions, rather than a transient afterhyperpolarization due to the antidromic spikes evoked by the optic nerve stimulation. Taken together, recurrent circuits from retinal ganglion cells are likely present widely across species. To gain full insights into these circuits' function, it will be critical next to unveil and compare the underlying synaptic mechanisms across species [43], especially if these ganglion-cell-coupled amacrine cells directly inhibit ganglion cells (postsynaptic inhibition) or instead inhibit bipolar cells that provide excitatory inputs to the ganglion cells (presynaptic inhibition) or both.

In this study, we employed an optic nerve stimulation to perturb ganglion cell activity independently of the light signaling pathway (Fig 4A). This successfully drove antidromic spikes in many ganglion cells in a time-locked manner (e.g., Fig 4B) and the obtained results provided direct experimental evidence supporting our model predictions on ganglion cell coupling properties. There are, however, some caveats, such as possible changes in the intrinsic properties of ganglion cells (Figs 5 and 6) and a potential activation of efferent axons from the brain to the retina [44, 45]. Previous studies suggest that dopaminergic amacrine cells receive such centrifugal inputs under the control of circadian rhythms [46]. Although dopamine can affect gap junction networks at nearly every stage of retinal processing, the kinetics are much slower than the time scales considered in this study (~100 ms) [47–49]. The effects of the efferent signaling, if any, should thus be negligible on the outcome of the nerve stimulation we observed. These dopaminergic amacrine cells are also suggested to receive direct inputs from intrinsically-photosensitive retinal ganglion cells via axon collaterals [50, 51]. For the same reason, however, the contribution of this dopaminergic pathway should also be minimal in this study.

Another caveat is that the optic nerve stimulation potentially led to an extensive perturbation of the coupling pathways due to a strong synchronous activation of ganglion cell populations. This may explain why the measured effect of the negative feedback lasted long, over hundreds of milliseconds (Figs 4, 5 and S5 Fig) as opposed to tens of milliseconds in the model

predictions (Fig 3 and S3 Fig). Single-cell stimulation, however, can be too weak to reveal any indirect negative interaction among ganglion cells (21, 24; but see 28, 29). It is a future challenge to further investigate the circuit mechanisms under more physiological conditions and test remaining model predictions, such as reciprocal negative couplings between retinal ganglion cells of the same response polarity (Fig 3G–3I and S3G–S3I Fig).

Recurrent normalization plays important roles in many brain functions [3]. Consistent with the previous study in the primate retina [27], both our computational model and experimental data show that the ganglion cell recurrent network plays a role in controlling the visual response gain in both salamander and mouse retinas (Fig 7 and S6 Fig). Interestingly, the interactions between ganglion cells extend over a wide spatial range (up to ~1 mm; Fig 3 and S3 Fig) well beyond the extent of their dendritic fields [18, 35]. Moreover, this gain control mechanism works on a slower time scale (~100 ms), if not the slowest [52], than many other local gain control mechanisms in the retina, such as the synaptic adaptation achieved at bipolar cell terminals at the cellular level [53, 54], or negative feedback loop among bipolar cells and amacrine cells at the circuit level [12–14]. Therefore, the ganglion cell feedback circuits likely contribute to slowly equalizing the response gain across populations, but not mediating visual functions that need precise spike timing of individual cells [55–58]. To investigate the structure and function of the feedback circuits for each ganglion cell type, future studies will benefit from further elaborations on the circuit models combined with calcium or voltage imaging techniques to monitor the subcellular activity of individual neurons and trace signal flow within the retina [59, 60].

## Materials and methods

No statistical method was used to predetermine the sample size. The significance level is 0.05 (with Bonferroni correction where appropriate) in all analyses unless noted otherwise. All experiments were performed in strict accordance with the protocols approved by the Institutional Animal Care and Use Committee at Harvard University or California Institute of Technology, or under the license 233/2017-PR from Italian Ministry of Health. The data analysis and circuit modeling were done in Matlab (Mathworks) and Python. All data and codes are available upon request.

### Modeling

We first reanalyzed the data sets in [30–32] for the salamander retinal ganglion cells and those in [33] for the mouse ganglion cells. Specifically, we focused on the responses to the random noise stimulus consisting of a 1-dimensional array of adjacent bars 8.3–80 μm in width. The light intensities of these bars were drawn from a binary black-or-white distribution (luminance range 0.5–36 mW/m2), and changed simultaneously, independently, and randomly with a refresh rate of 60–100 Hz.

The distance between the cells (Fig 3 and S3 Fig) was calculated from the receptive field centers. The spatiotemporal receptive fields of the cells (10–17 ms bin size; 0.4 s window) were estimated by reverse-correlation methods using randomly flickering checkerboard stimuli (30–83 μm square fields; 60–100 frames/s), and their center location was estimated by the 3-dimensional Gaussian curve fit. Cells with positive receptive field center values were classified as ON cells, whereas those with negative values as OFF cells.

**Data selection.** The raw data sets contained 466 OFF and 13 ON ganglion cells from 10 isolated salamander retinas, and 22 OFF and 13 ON ganglion cells from a mouse retina. Of those, 185 OFF and 4 ON salamander cells and 10 ON and 12 OFF mouse cells were selected for the subsequent modeling analyses according to the following predetermined criteria.

1. Cells should have a high spike sorting quality. To ensure low false positives, cells with >15% of spikes with <1.7 ms inter-spike intervals were discarded. To ensure low false negatives, cells with nearly identical spatiotemporal receptive field profiles were eliminated except for the one with the highest spike counts as we cannot exclude the possibility that they were the same cell.

2. Cells should respond well to the visual stimulus for robust model fitting, assessed by the spike counts during the stimulus presentation period and the performance of a linear-non-linear (LN) cascade model [40, 41]. The LN model was fitted by reverse-correlation methods, and cells with <3,000 spikes and <10% LN model prediction (coefficient of determination; see below Eq (3)) were discarded.

3. Cells should have little response correlation due to common visual inputs for modelling interactions between cells. Neighboring cells were included in the coupling models (see below Model formalism) only if Pearson correlation between their spike trains and those of the target cell was low (between −0.1 and 0.1; S2A and S2B Fig). This generally resulted in no overlap of the receptive field centers (e.g., Fig 1B).

**Model formalism.** We employed the cascade model framework as described in [30]. As a non-coupled model (Figs 1, 2 and S1 and S4 Figs), we thus used the so-called "linear-nonlinear-feedback-delayed-sum-nonlinear-feedback" (LNFDSNF) or full model, and the "linear-nonlinear-sum-nonlinear" (LNSN) or reduced model. In short, the first LNF stages collectively work as spatial subunits of upstream bipolar cells (Bipolar Cell Modules; Eqs (S3)–(S7) in [30]); the middle D is a delay due to lateral signal propagation via amacrine cells (Amacrine Cell Modules; Eq (S8) in [30]); and the last SNF stages represent the spatial summation by a target ganglion cell (Ganglion Cell Modules; GCM; Eq (S9) in [30]), followed by the cell's output nonlinearity (Eq (S6') below) and the feedback (Eq (S7) in [30]). Here we replaced the Eq (S6) in [30] with the following formula:

$$y(t) = \begin{cases} 0, & \text{if } z(t) \leq \theta \\ \alpha(z(t) - \theta), & \text{otherwise} \end{cases} \tag{S6}$$

where $\alpha$ and $\theta$ are the slope and threshold of a half-wave rectification function, respectively. The GCM nonlinearity was thus equipped with two free parameters in this study (S4 Fig), while it was fixed in [30], equivalent to $\alpha = 1$ and $\theta = 0$ in Eq (S6').

A full coupling model (LNFDSCNF; S1A Fig) and a reduced one (LNSCN; Fig 1A) were then built by introducing the coupling step (C) before the GCM nonlinearity (N) and feedback (F). Specifically, the coupling effects on a target ganglion cell were modelled as a delayed weighted sum of the neighboring ganglion cell activities. For each $k$-th neighboring cell, two free parameters were hence assigned: $l_k$ for the latency of the signal transmission and $a_k$ for the coupling strength. The latency parameter $l_k$ is non-negative and treated as a delay function that requires interpolation of the measured firing rate $r_k(t)$ of the $k$-th neighboring cell. The delayed activity $r_k^*(t)$ is thus given as:

$$r_k^*(t) = (1 - \{l_k\})r_k(t - \lfloor l_k \rfloor) + \{l_k\}r_k(t - \lfloor l_k \rfloor - 1), \tag{1}$$

where $\lfloor l_k \rfloor$ is the largest integer not greater than $l_k$, and the fractional part $\{l_k\} = l_k - \lfloor l_k \rfloor$. The delayed activity $r_k^*(t)$ is then weighted by a coupling strength $a_k$ and added to the signal $x(t)$

coming from the previous summation (S) step (Eq (S9) in [30]):

$$y(t) = x(t) + \sum_k a_k r_k^*(t). \tag{2}$$

The resulting signal $y(t)$ is in turn used as an input to the following GCM nonlinearity (Eq (S6')) and feedback steps (Eq (S7) in [30]) to obtain an estimated firing rate of the target cell (e.g., Fig 1).

**Model fitting and assessment.**   We wrote custom codes in Python to fit the models to the ganglion cell firing rates (bin size, 1/100–1/60 s) in response to the randomly flickering bar stimuli. Specifically, the "optimize.minimize" function from SciPy library was used to minimize the objective function represented by the mean squared error between the measured firing rate and the model output. For each cell, we fitted the coupled models in two configurations (Figs 1–3 and S1–S4 Figs): the one including only those surrounding cells of the same response polarity as the target cell (e.g., OFF target cell with OFF surrounding cells), and the other with those of different response polarities alone (e.g., OFF target cell with ON surrounding cells). The Kolmogorov-Smirnov test was performed to compare the model performance gain over the non-coupling models (see below) in the two configurations (Fig 2 and S1B–S1E Fig).

Model performance was assessed by the coefficient of determination between the measured ganglion cell firing rate $r(t)$ and the model prediction $\hat{r}(t)$:

$$R^2 = 1 - \frac{\sum_t (r(t) - \hat{r}(t))^2}{\sum_t (r(t) - \langle r(t) \rangle)^2}, \tag{3}$$

where $\langle \cdot \rangle$ denotes the mean. It reaches its maximum of 1 in the case of an exact agreement between the two binned sequences, and is around 0 or less in the case of unrelated sequences. Paired t-test was used to compare the model performance with and without couplings (Fig 2 and S1B–S1E Fig). Because the coupled model did not improve its performance for mouse cells with surrounding cells of different response polarities, this data set was excluded from the subsequent model parameter analyses (Fig 3 and S2–S4 Figs).

To avoid over-fitting, the model parameters were optimized using a training data set (~80% of the data) and the model performance was evaluated on a separate testing data set. Testing data set for some salamander cells included 8–12 repeats of the identical flicker sequence [30]. In these cases, the model's output was compared to the average firing rate over all these trials, while the model was trained with a separate training data set.

**Model analysis.**   We ran a shuffling analysis to evaluate the noise level of the coupling parameters in the LNSCN and LNFDSCNF models (lower and upper thresholds at 0.5 and 99.5 percentiles, respectively; Fig 3 and S2 and S3 Figs). Specifically, for each modelled cell, we fitted the model parameters to its true spike trains together with randomly jittered spikes trains from all coupled neighboring cells. This keeps intact the stimulus-dependence of the modelled cell's response, but breaks the correlation to the responses of the other cells. Thus, the obtained coupling parameters form a distribution expected from a chance level.

We performed a perturbation analysis to test the convergence and sensitivity of the model parameters. Specifically, after fitting LNSCN and LNFDSCNF models, we reset the coupling parameters to be zero while keeping the other parameters intact, and refitted the models to obtain a new set of the parameters. We then compared the original and reoptimized coupling strength and delay parameters using Pearson's correlation coefficient (S2C and S2D Fig). The difference of the model performance ($R^2$ in Eq (3)) between the original and reoptimized models was used to determine the confidence interval of the performance gain from the non-coupling to the coupling models (lower and upper thresholds at 5 and 95 percentiles, respectively; Fig 2 and S1B–S1E Fig).

For the population analysis of the coupling properties (Fig 3 and S3 Fig), we first pulled together the parameter values for each cell pair across all coupling models. We then performed the Wilcoxon rank-sum test to examine the distance and the signal delay between the positively and negatively coupled cells (Fig 3A–3F and S3A–S3F Fig). To analyze the symmetry of the coupling effects between cells, we examined by a $\chi^2$-test if the polarity of the signal from one cell to another depends on that in the opposite direction (*df* = 4; Fig 3G–3I and S3G–S3I Fig). The data sets from ON and OFF cells were combined as they showed the same trend.

We analyzed in three ways how the coupling affects the behavior of the ganglion cell module (GCM) in the circuit models (S4 Fig). First, we assessed the input dynamics to GCM as a collective measure of the model's upstream circuit properties. Specifically, for each cell, we compared the outputs of the summation (S) stage in the coupled and non-coupled models [Eq (S9) in 30] using Pearson's correlation coefficient (S4A and S4D Fig). Second, we examined the GCM feedback filters from the parameters of the second feedback (F) stage in the full models (Eq (S7) in [30]; S4B and S4E Fig). Finally, we compared the GCM nonlinearity parameters (threshold $\theta$ and slope $\alpha$ for the second nonlinearity (N) stage; Eq (S6')) using Pearson's correlation coefficient between the coupled and non-coupled models (S4C and S4F Fig).

## Electrophysiology

The dark-adapted retina of a larval tiger salamander (*Ambystoma tigrinum*) or an adult wild-type mouse (*Mus musculus*; C57BL/6J strain, 2–6 months old) was isolated with an intact optic nerve attached (~5 mm long), and placed on a flat array of 61 extracellular electrodes with the ganglion cell side down. The salamander retina was superfused with oxygenated Ringer's medium (in mM: NaCl, 110; NaHCO$_3$, 22; KCl, 2.5; MgCl$_2$, 1.6; CaCl$_2$, 1; and D-glucose, 10; equilibrated with 95% O$_2$ and 5% CO$_2$ gas) at room temperature, and the mouse retina with oxygenated Ames' medium (Sigma-Aldrich, A1420) at 37˚C. The electrode array recorded the extracellular signals from ganglion cells with each electrode sampled at 10 kHz, while photoreceptors and/or the optic nerve were stimulated visually and/or electrically, respectively. Spike trains from individual ganglion cells were extracted from raw voltage traces by a semiautomated spike-sorting algorithm [61] written in IGOR Pro (Wave Metrics).

We adapted the methods described in [38] to electrically stimulate the tip of the optic nerve emerging from an isolated retina using a suction electrode (Fig 4A). Specifically, we used bipolar pulses (10–50 V, 0.02–0.5 ms) from a stimulus isolator (Grass Instrument, SD9) at 2/3–1 Hz (100–200 trials in the dark; 800–2,000 trials with visual stimulation) controlled by custom software written in LabView (National Instruments). Antidromically evoked spikes were observed in some ganglion cells at a latency of around 5 ms after the nerve stimulation (e.g., Fig 4B and 4C and S5A and S5B Fig). Due to the stimulus artifacts, we were not able to detect any spikes within a few milliseconds after the nerve stimulation (e.g., Fig 4B). When the suction electrode was inserted into the bath without holding the nerve ending, the electrical stimulation did not affect the spontaneous activity of ganglion cells (3 salamander and 2 mouse retinas).

In total, recordings were made from 855 ganglion cells in 19 salamander retinas and 368 cells in 7 mouse retinas for the optic nerve stimulation experiments. Of those, all salamander retinas were examined with a high-potassium Ringer's medium (13 mM KCl replacing NaCl; Fig 5B), 349 cells in 11 salamander retinas with a gap junction blocker (100 μM meclofenamic acid; Fig 6A and 6B), 380 cells in 6 salamander retina with inhibitory synaptic transmission blockers (100 μM picrotoxin and 1.0 μM strychnine; Fig 6C and 6D), and 160 cells in 6 salamander retinas and 172 cells in 3 mouse retinas with visual stimulation (Fig 7 and S6 Fig). As we focused on a suppression of ganglion cell activity, only those cells with a sufficiently high baseline firing rate (>1 Hz) were selected for subsequent analyses.

## Visual stimulation

Visual stimuli were displayed on a gamma-corrected cathode-ray tube monitor (DELL E773c; frame rate 100 Hz; mean luminance 18 mW/m$^2$) and projected onto the photoreceptor layer of the retina from above through a custom-made lens system. We presented full-field random flicker stimuli (100 frames per second; light intensities drawn from Gaussian distribution with mean luminance of 18 mW/m$^2$ and standard deviation of 7 mW/m$^2$) for examining the effects of the optic nerve stimulation on the ganglion cell visual responses (Fig 7 and S6 Fig). Here we could not use spatially-structured stimuli because the stimulation pipette created a distortion in an uncontrollable manner.

## Data analysis

**Optic nerve stimulation in the dark.** To measure the effects of the optic nerve stimulation in the dark (Figs 4–6 and S5 Fig), we first computed the peri-stimulus time histogram (PSTH) for each ganglion cell with increasing bin sizes (0–10, 10–20, 20–40, 40–80, 80–160, 160–320, and 320–640 ms after the nerve stimulation), and identified those time bins that had significantly different firing rates from the baseline activity (320 ms period before the onset of the nerve stimulation) using bootstrap resampling methods over trials (10,000 repeats). Here we chose unequal bin sizes 1) to robustly assess a decrease of the firing rate even for those cells with relatively low baseline firing rates (threshold at 1 Hz); and 2) to avoid the problem of multiple comparisons with respect to the number of time bins for testing the nerve stimulation effect in each cell. Cells with a significantly increased firing rate in the first time bin (0–10 ms) were considered as the ones directly evoked by the nerve stimulation (Fig 5A, 5B and S5C Fig, top rows) because antidromic spikes typically had a latency of around 5 ms (e.g., Fig 4B and 4C and S5A and S5B Fig). Cells with significantly different firing rates in the second time bins and thereafter (i.e., >10 ms latency) were considered as the ones indirectly affected by the nerve stimulation (Fig 5A and 5B and S5C Fig, bottom rows). We ran a $\chi^2$-test to compare the proportions of the cells in the normal (Fig 5A) and high-potassium (Fig 5B) conditions that showed antidromic spikes, enhanced firing, or suppressed firing after the nerve stimulation, respectively. For a display purpose, we computed the PSTHs with 20 ms bins in Fig 4C and 4D and S5A and S5B Fig, and labeled those significantly below or above the baseline (250 ms period before the onset of the nerve stimulation) in blue and red shades, respectively, without correcting the significance level by the number of time bins.

**Optic nerve stimulation with visual stimulation.** We used stimulus ensemble statistical techniques ("reverse correlation" methods; 400 ms window; 10 ms bin width) to calculate the linear filter and static nonlinear gain function of the recorded cells (Fig 7 and S6 Fig). Specifically, for each ganglion cell, we first computed the PSTH (100 ms bin width) with respect to the nerve stimulation during the full-field random flicker stimulus presentation, and identified those time bins with significantly higher or lower firing rates than the baseline (500 ms period before the nerve stimulation) using the bootstrap resampling methods over trials (10,000 repeats; e.g., Fig 7B and S6B Fig). For each time bin, we then estimated the linear filter by a spike-triggered average stimulus (e.g., Fig 7C and S6C Fig). The obtained linear filters indicate the average stimulus features that made the cell fire action potentials. We then characterized the change in their profile from the baseline in two ways. First, we used the ON-OFF index, defined as the difference between the peak and valley values of the linear filter, normalized by the sum of the two:

$$\text{ON}-\text{OFF index} = \frac{|\text{peak}| - |\text{valley}|}{|\text{peak}| + |\text{valley}|}. \tag{4}$$

The ON-OFF index value of −1, 1, and 0 indicates the stimulus feature selectivity towards purely OFF stimuli, purely ON stimuli, and both ON and OFF stimuli, respectively (e.g., Fig 7D and S6D Fig). Second, we calculated the power spectral density of the linear filter and estimated the peak frequency by fitting a gaussian curve as a measure of the cell's temporal frequency tuning (e.g., Fig 7E and 7F and S6E and S6F Fig). Sign-test was used to evaluate the change in these properties over the population during the suppressed and enhanced firing periods (Fig 7I and 7J and S6I and S6J Fig).

For each epoch, we also computed the static nonlinear gain function $P(\text{response}|\text{stimulus})$ as a function of the linear filter output values by the convolution with stimulus fragments:

$$P(\text{response}|\text{stimulus}) = \frac{N(\text{stimulus}|\text{response})}{N(\text{stimulus})}, \tag{5}$$

where $N(\text{stimulus})$ and $N(\text{stimulus}|\text{response})$ are the distributions obtained from all stimuli and spike-triggered stimulus ensembles, respectively (e.g., Fig 7G and S6G Fig). For quantification, we fitted the following sigmoid function $f(x)$ to the gain function profiles:

$$f(x) = \frac{u - l}{1 + \exp[-s(x - c)]} + l, \tag{6}$$

and calculated the change in the parameters ($u$, upper bound; $l$, lower bound; $c$, center; $s$, slope) from the baseline (e.g., Fig 7H and S6H Fig). Sign-test was performed to test the statistical significance (Fig 7K and 7L and S6K and S6L Fig).

## Supporting information

**S1 Fig. Retinal ganglion cell couplings improve the performance of the full circuit model.** (A) Schematic diagram of the full circuit model with couplings ("LNFDSCNF" model). The model without couplings is identical to the "LNFDSNF" model in [30]. (B–E) Cumulative probability distributions (black; confidence intervals in gray) of LNFDSCNF model performance gain over LNFDSNF model for salamander (B, $\Delta R^2$ = 0.008±0.008 (mean ± standard deviation), $p<0.001$ (paired t-test), for coupled cells with the same response polarity; C, $\Delta R^2$ = 0.005±0.007, $p<0.001$, for those with different polarities; $p = 0.054$ from Kolmogorov-Smirnov test) and mouse (D, $\Delta R^2$ = 0.004±0.002, $p<0.001$; E, $\Delta R2$ = 0.000±0.001, $p = 0.4$; $p<0.001$ from Kolmogorov-Smirnov test) retinal ganglion cells. The data are shown in the same format as Fig 2. (F, G) Time course of the firing rate (red) of representative salamander (F; cell #59 in Fig 1) and mouse (G) cells and that of the model outputs with (black, LNFDSCNF model) and without (cyan, LNFDSNF model) couplings. Arrows indicate the false-positive responses in the LNFDSNF model that were correctly suppressed in the LNFDSCNF model. (PDF)

**S2 Fig. Ganglion cell pairs with low spike correlations can have strong couplings in either polarity.** (A) Pairwise spike correlations of the representative cell (#25 from Fig 1B–1D; auto-correlation) and all the other simultaneously recorded 19 cells (cross-correlations). Those cells with too high correlations (gray; ≥0.1 or ≤−0.1 at the peak) were excluded from our model analysis to minimize the confounding effects of common visual inputs (see Methods for details). The coupling strength is shown on the bottom-right for each cell included in the LNSCN model targeting the cell #25 (black). (B) Coupling strength from the LNSCN (top) and LNFDSCNF (bottom) models as a function of the Pearson cross-correlation of firing patterns between ganglion cell pairs (left, salamander; right, mouse). Although we selected only those cells with low spike correlations (from −0.1 to 0.1) in the coupling models, we frequently found strong couplings in either polarity. The cells in the representative data set (from A) are

highlighted with yellow circles. (C, D) Comparison of the model parameters (C, coupling strength; D, delay) between the original and reoptimized models after resetting them to be zero (top, LNSCN; bottom, LNFDSCNF; left, salamander; right, mouse). High Pearson cross-correlation values (*R*) in the perturbation analysis support a good convergence of the model fitting (see Methods for details).
(PDF)

**S3 Fig. The circuit model predictions hold with the full circuit model.** The data are shown in the same format as Fig 3 but for the LNFDSCNF model. (A, B) Coupling strength (A; $a_k$ in Eq (2) in Methods) and delay (B; $l_k$ in Eq (1) in Methods) parameter values plotted as a function of the distance between salamander retinal ganglion cells of the same response polarity. As is the case with the LNSCN model (Fig 3), positive couplings (red, *N* = 154) were found at a shorter distance (A; 0.31±0.14 mm versus 0.36±0.20 mm; median ± interquartile range (box plot); *p*<0.001, rank-sum test) and with a shorter latency (B; 2.3±4.7 versus 4.7±5.2 ms; *p*<0.001) than negative couplings (blue, *N* = 64). (C, D) Corresponding figure panels for salamander cells with different response polarities. Couplings above the noise level were all positive (*N* = 11) with the distance of 0.29±0.06 mm (C; median ± interquartile range) and the latency of 3.8±6.5 ms (D). (E, F) Corresponding figure panels for mouse retinal ganglion cells of the same response polarity. Likewise, positive couplings (red, *N* = 37) were found at a shorter distance (C; 0.35±0.28 mm versus 0.72±0.65 mm; *p* = 0.02) and with a shorter latency (D; 0.00±0.04 ms versus 19.1±7.7 ms; *p*<0.001) than negative couplings (blue, *N* = 6). Data not analyzed for the mouse cells with different response polarities because the coupled model performance did not improve significantly (S1E Fig). (G–I) Comparison of the coupling strengths from one cell to another and vice versa. Ganglion cells have symmetric couplings more frequently than expected in both salamander (G, *p*<0.001, $\chi^2$-test with *df* = 4, between cells of the same response polarity; H, *p*<0.001, between cells with different response polarities) and mouse (I; *p*<0.001, between cells of the same response polarity) retinas. The number of data points in each category is shown in the figure panels.
(PDF)

**S4 Fig. Ganglion cell module properties do not differ between the coupled and non-coupled models.** (A) Distribution of the ganglion cell module (GCM) input correlation between coupled and non-coupled models fitted to salamander cells (black, reduced models: LNSCN versus LNSN, 0.987±0.020, median ± interquartile range; gray, full models: LNFDSCNF versus LNFDSNF, 0.996±0.005). The correlation was calculated using the outputs of the summation (S) stage in the models (Eq (S9) in [30]). High correlations indicate that the GCM input dynamics are nearly identical in response to the visual stimuli (arrow, representative cell in Fig 1B–1D). (B) The second feedback (F) stage (GCM feedback filters; Eq (S7) in [30]) was nearly identical between the non-coupled (left, LNFDSNF) and the coupled (right, LNFDSCNF) models (yellow, representative cell in Fig 1B–1D; gray, all the other cells; red, mean). (C) Comparison of the second nonlinear (N) stage (GCM nonlinearity) between the coupled and non-coupled models (top, LNSN versus LNSCN; bottom, LNFDSNF versus LNFDSCNF; yellow, representative cell in Fig 1B–1D. Couplings did not affect the slope (left, $\alpha$ in Eq (S6'); Pearson's *R* = 0.979 and 0.997 for the reduced and full models, respectively) or the threshold (right, $\theta$ in Eq (S6'); *R* = 0.955 and 0.997, respectively). (D–F) Corresponding figure panels for mouse retinal ganglion cells. There was no marked difference between the coupled and non-coupled models in GCM input correlations (D; black, reduced models, 0.994±0.005; gray, full models, 0.998 ±0.003), GCM feedback (E), GCM slope (F, left; *R* = 0.967 and 0.978 for the reduced (top) and full (bottom) models, respectively) or GCM slope (F, right; *R* = 0.991 and 0.996, respectively).
(PDF)

**S5 Fig. Optic nerve stimulation produces fast excitation and slow inhibition in ganglion cell firing activity in isolated mouse retinas.** The data are shown in the same format as Figs 4 and 5 but for the mouse retina. (A, B) Firing patterns of two representative retinal ganglion cells, showing a period of suppression after the optic nerve stimulation (asterisk, antidromically evoked spikes) in the dark. Note the cell in B showed a period of enhanced firing before the suppression, while the one in A did not. (C) Population data of the ganglion cell responses to the nerve stimulation in the format of 2-by-4 contingency table, categorized by direct effects in rows and indirect effects in columns. Top row, cells with antidromic spikes; bottom row, cells without antidromic spikes. First column, no indirect effect; second and third columns, enhanced firing after the nerve stimulation (red circles, bins with significantly increased firing from the baseline); third and fourth columns, suppressed firing after the nerve stimulation (blue circles, significant decrease).
(PDF)

**S6 Fig. Negative feedback modulates the visual response properties of mouse retinal ganglion cells.** The data are shown in the same format as Fig 7 but for the mouse retina. (A) Schematic diagram of the experiment and analysis. (B) PSTH of a representative mouse ganglion cell with respect to the optic nerve stimulation during the visual stimulus presentation. Blue- and red-shaded bins indicate those with significantly higher and lower firing rates than the baseline, respectively. (C, D) Linear filters of the example cell (left) and corresponding ON-OFF indices (D; Eq (4) in Methods) at different time bins from the nerve stimulation. The filter became more biphasic during which the visual responses were enhanced by the nerve stimulation (red, 0–100 ms bin with significantly higher firing rates), but it went back to normal even when the visual responses were suppressed (blue, 200–300 ms bin with significantly lower firing rates; gray, all the other 100 ms bins, with the mean in black). (E, F) Normalized power spectral density (E) and the peak frequency (F; estimated by a Gaussian curve fit) of the linear filters in C. (G, H) Static nonlinearities of the example cell (G; Eq (5) in Methods) and the upper and lower bounds of the cell's firing probability from sigmoidal curve fits (H; Eq (6) in Methods) at different time bins from the nerve stimulation. The upper bound was lower during the suppressed firing period (blue, 200–300 ms bin) while the lower bound was higher during the enhanced firing period (red, 0–100 ms bin) compared to the control period (gray, all the other 100 ms bins; black, mean). (I, J) Summary of the changes in the ON-OFF indices (I; suppressed, $-0.09\pm0.08$; enhanced, $0.12\pm0.21$) and the peak frequencies (J; $0.38\pm0.83$ and $0.26\pm2.29$) between the periods with and without significant firing rate changes after the nerve stimulation (blue, decrease $N = 4$; red, increase $N = 17$; magenta, example cell in B–H). (K, L) Summary of the changes in the sigmoid function parameters fitted to the nonlinearity (K, upper bounds, $-0.04\pm0.05$ and $-0.00\pm0.03$; L, lower bounds, $-0.00\pm0.01$ and $0.01\pm0.02$; magenta, example cell in B–H).
(PDF)

## Acknowledgments

We gratefully acknowledge Markus Meister for assistance with experiments. We also thank all the members of the Asari lab at EMBL Rome as well as Cornelius Gross and Santiago Rompani for many useful discussions.

## Author Contributions

**Conceptualization:** Anastasiia Vlasiuk, Hiroki Asari.

**Data curation:** Anastasiia Vlasiuk, Hiroki Asari.

**Formal analysis:** Anastasiia Vlasiuk, Hiroki Asari.

**Investigation:** Anastasiia Vlasiuk, Hiroki Asari.

**Methodology:** Anastasiia Vlasiuk, Hiroki Asari.

**Supervision:** Hiroki Asari.

**Writing – original draft:** Anastasiia Vlasiuk, Hiroki Asari.

**Writing – review & editing:** Anastasiia Vlasiuk, Hiroki Asari.

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
