## [Decision Letter · Decision Letter 0]

28 Apr 2021

PONE-D-21-08919

Feedback from retinal ganglion cells to the inner retina

PLOS ONE

Dear Dr. Asari,

Thank you for submitting your manuscript to PLOS ONE. After careful consideration, we feel that it has merit but does not fully meet PLOS ONE’s publication criteria as it currently stands. Therefore, we invite you to submit a revised version of the manuscript that addresses the points raised during the review process.

The two expert reviewers have provided many suggestions to improve the manuscript's integrity, many of which are deemed rather important. Please make changes, corrections or extended explanations in every case.

We look forward to receiving your revised manuscript.

Kind regards,

Steven Barnes

Academic Editor

PLOS ONE

Journal Requirements:

Reviewers' comments:

Reviewer's Responses to Questions

**Comments to the Author**

1. Is the manuscript technically sound, and do the data support the conclusions?

Reviewer #1: Yes

Reviewer #2: Yes

2. Has the statistical analysis been performed appropriately and rigorously? 

Reviewer #1: Yes

Reviewer #2: Yes

3. Have the authors made all data underlying the findings in their manuscript fully available?

Reviewer #1: Yes

Reviewer #2: Yes

4. Is the manuscript presented in an intelligible fashion and written in standard English?

Reviewer #1: Yes

Reviewer #2: Yes

5. Review Comments to the Author

Reviewer #1: Review: Feedback from retinal ganglion cells to the inner retina

Summary

The authors use a combination of circuit modeling and experimental approaches to show that two distinct coupling pathways mediate local feedback from retinal ganglion cells in the retina. These two pathways consist of: (1) a fast and positive coupling between proximal ganglion cells and, (2) a slower negative coupling between distal ganglion cells via the connections with amacrine cells. These coupling pathways modulates the retinal ganglion cells’ response gain without affecting their feature selectivity.

The authors apply electrical perturbation of optic nerve and simultaneous use multi-electrode arrays to record the spiking activity from either salamander or mouse retinas. These recordings showed responses consistent with the existence of feedback circuity from retinal ganglion cells. Optic nerve stimulation triggered two distinct firing patterns of retinal ganglion cells: (1) a fast enhancement and (2) a slow, delayed suppression of spiking activity. Using different concentrations of potassium to modulate background firing rate they show that increasing spontaneous activity of retinal ganglion cells only affect the fast enhancement. This result supports the claim that the positive coupling is based on direct connections between retinal ganglion cells. The slow suppression could be eliminated by either blocking gap junctions or by blocking the inhibitory transmitters of amacrine cells. These results support the authors’ claim that the negative coupling pathway depends on the connections between retinal ganglion cells and amacrine cells. These findings are consistent with their modeling predictions.

Subsequently, to investigate how these coupling pathways affect the visual response properties of retinal ganglion cells, the authors record the ganglion cells’ visual response to a white-noise stimulus in combination with the optic nerve stimulation. The authors found that the ON-OFF index of each ganglion cell, which represent the feature selectivity remained largely unchanged. In contrast, the response gain is down-regulated during a period of suppression and up-regulated during a period of enhancement. These results support their claims that the feedback signaling pathway from ganglion cells could modulates the visual response gain but not the feature selectivity of retinal ganglion cells.

Taken together, we support publishing this paper with a major revision and some minor revisions stated below:

Major

First, the authors should add the comparison between the LNFDSNF model and the LNFDSCNF model. Figure 1D only shows that the LNSCN can explain the false-positive derived from the LNSN models. However, it is not clear whether this false positive could also be eliminated by just using the full model of LNFDSNF without considering the coupling parameter but only the feedback delay parameters. This needs to be clearly stipulated in the results section and ideally included in Figure 1D.

Minor

Line 78-79: ‘~10s ms’ should be ~10 ms. ‘~100s ms’ should be ~100 ms.

Line 128-131: Delete ‘much’. The differences are that different from above where you use the word small.

Line 498 & Figure 3: The authors should clarify whether the model cell is an ON, OFF, or ON-OFF cell. The authors should also clarify the number of cells used in the analysis.

Line 222 & Figure5A-B: The number of the ganglion cells used for data analysis is not clear. For example, the cells are separated into three groups: cells with only enhanced firing (23 cells), cells with only suppression of firing (13 cells), and cells with both enhanced firing and suppression (5 cells). And the total number is 34, how different cells are assigned to each group should be clarified.

Line 244 & 253: The authors claim that blocking gap junctions ‘had little effect on the fast enhancement of the firing’ and blocking inhibitory transmission leave ‘the fast enhancement intact’. Is this conclusion based on the fraction of the cells with enhancement, or the firing activity of the same cell with enhancement in each condition? Please clarify in the result section.

Line 266: ‘fitted a linear-nonlinear (LN) cascade model to the visual responses of each cell’? The authors should explain which LN, LNSCN, or LNSN model is used here and why.

Reviewer #2: In their paper, the authors investigate how RGCs can influence each other via gap junctions. To this end, they use data from salamander and mouse retina and first fit models to the data, which demonstrates statistical couplings between RGCs. They then stimulate the optic nerve to induce activity in RGCs, and find inhibitory slow effects, which they attribute to ACs based on pharmacological manipulation.

Overall, the paper is clearly written and appears technically correct. I have a few suggestions/remarks:

1) Line 150: The authors claim that couplings were typically reciprocal, but I could not find a quantification of this finding.

2) Line 151: I don’t understand what evidence leads the authors to conclude that “single RGCs can be involved in multiple feedback pathways”.

3) Line 198ff: Can the authors estimate how many RGCs are stimulated by their optic nerve manipulation?

4) Fig. 5A: The suppressive effects seem very small (I understand that they are hard to quantify). Even circles effectively showing zero effect are labeled significant. How was the test done? Is there some issue here?

5) Line 215: Please show all data.

6) Lines 257 ff: I am surprised that the fitting for these very short periods after the nerve shock works so well. How many spikes went into this analysis for single cells in the respective periods?

7) Line 414: I am aware of an OFF bias in salamander recordings, but 185 to 4 seems excessive. What happened to the ON cells?

8) Line 418: How does the selection of cells chosen by the authors ensure low false negatives? Were these criteria determined before the analysis? How are the cells without clear LN model fit affected?

9) Lines 461: What optimizer at which settings was used to fit the model? Is the code available online?

10) Line 479: If the stimulus sequence during the test trials was identical to the training trials, the authors use a very weak notion of generalization. Could the authors also report generalization to data unseen during training? E.g. perform cross-validation with hold-out stimulus sequences.

11) Line 566: Unclear which multiple comparison problem is being referred to. Consider ANOVA with post-hoc testing.

6. PLOS authors have the option to publish the peer review history of their article (what does this mean?). If published, this will include your full peer review and any attached files.

Reviewer #1: No

Reviewer #2: No

---

## [Author Response · Author response to Decision Letter 0]

5 Jun 2021

--- journal requirements ---

1. We have revised our manuscript in the PLOS ONE's style format.

2. We confirmed that the reference list is complete and correct.

3. We have included all relevant data in the revised manuscript. See also our response to Rev#2, remark 5). 

--- reviewers' comments:

We have addressed all the concerns by the two reviewers. Please see enclosed our response to reviewers for details.

---

## [Decision Letter · Decision Letter 1]

30 Jun 2021

Feedback from retinal ganglion cells to the inner retina

PONE-D-21-08919R1

Dear Dr. Asari,

We’re pleased to inform you that your manuscript has been judged scientifically suitable for publication and will be formally accepted for publication once it meets all outstanding technical requirements.

Kind regards,

Steven Barnes

Academic Editor

PLOS ONE

Additional Editor Comments (optional):

Due to Reviewer #2 being unable to assess the authors' responses to their concerns, I took it upon myself to carefully consider the merit of the responses and pass judgement in the place of Reviewer #2. The concerns raised by Reviewer #2 had mostly addressed required clarifications of the data and their explanation as presented by the authors. Reviewer #2 had considered these to be "Minor" corrections and I carefully assessed each of the responses to Reviewer #2's concerns in the Revision submitted by the authors. I found each and every concern raised by Reviewer #2 to be adequately satisfied with the corrections in the authors' revisions to the manuscript.

On this basis, together with the conclusion arrived at by Reviewer #1, I determined that this manuscript now meets the requirements for publication in PLOS ONE.

Reviewers' comments:

Reviewer's Responses to Questions

**Comments to the Author**

1. If the authors have adequately addressed your comments raised in a previous round of review and you feel that this manuscript is now acceptable for publication, you may indicate that here to bypass the “Comments to the Author” section, enter your conflict of interest statement in the “Confidential to Editor” section, and submit your "Accept" recommendation.

Reviewer #1: All comments have been addressed

2. Is the manuscript technically sound, and do the data support the conclusions?

Reviewer #1: Yes

3. Has the statistical analysis been performed appropriately and rigorously? 

Reviewer #1: Yes

4. Have the authors made all data underlying the findings in their manuscript fully available?

Reviewer #1: Yes

5. Is the manuscript presented in an intelligible fashion and written in standard English?

Reviewer #1: Yes

6. Review Comments to the Author

Reviewer #1: In the revised manuscript, the authors have addressed all the major and the minor revisions. The authors added the comparison between the LNFDSNF model and the LNFDSCNF mode (in supplementary Figure 1F-G). The result showed the false-positive responses in the LNFDSNF model can be eliminated in the LNFDSCNF model. It explained our concerns: 1) the feedback delay parameter is not enough to eliminate the false-positive responses; 2) including the coupling parameter correctly suppress the false-positive responses. Additionally, the authors have clarified the number of cells used for different analysis and the statistical result to support their conclusion. These editions addressed our questions. Taken together, I support publishing this paper with no more revisions.

7. PLOS authors have the option to publish the peer review history of their article (what does this mean?). If published, this will include your full peer review and any attached files.

Reviewer #1: No

---

## [Editor Report · Acceptance letter]

12 Jul 2021

PONE-D-21-08919R1 

Feedback from retinal ganglion cells to the inner retina 

Dear Dr. Asari:

I'm pleased to inform you that your manuscript has been deemed suitable for publication in PLOS ONE. Congratulations! Your manuscript is now with our production department. 

Kind regards, 

on behalf of

Dr. Steven Barnes 

Academic Editor

PLOS ONE